# High-Precision CO_2_ Column Length Analysis on the Basis of a 1.57-μm Dual-Wavelength IPDA Lidar

**DOI:** 10.3390/s20205887

**Published:** 2020-10-17

**Authors:** Xin Ma, Haowei Zhang, Ge Han, Hao Xu, Tianqi Shi, Wei Gong, Yue Ma, Song Li

**Affiliations:** 1State Key Laboratory of Information Engineering in Surveying, Mapping and Remote Sensing, Wuhan University, Wuhan 473079, China; maxinwhu@whu.edu.cn (X.M.); xiaohao190081@whu.edu.cn (H.X.); shitian@whu.edu.cn (T.S.); weigong@whu.edu.cn (W.G.); 2CAS Key Laboratory of Spectral Imaging Technology, Xi’an 710119, China; 3School of Remote Sensing and Information Engineering, Wuhan University, Wuhan 473079, China; udhan@whu.edu.cn; 4School of Electronic Information, Wuhan University, Wuhan 473072, China; yue.ma3@adfa.edu.au (Y.M.); ls@whu.edu.cn (S.L.)

**Keywords:** IPDA Lidar, CO_2_, ranging, dual-wavelength

## Abstract

For high-precision measurements of the CO_2_ column concentration in the atmosphere with airborne integrated path differential absorption (IPDA) Lidar, the exact distance of the Lidar beam to the scattering surface, that is, the length of the column, must be measured accurately. For the high-precision inversion of the column length, we propose a set of methods on the basis of the actual conditions, including autocorrelation detection, adaptive filtering, Gaussian decomposition, and optimized Levenberg–Marquardt fitting based on the generalized Gaussian distribution. Then, based on the information of a pair of laser pulses, we use the direct adjustment method of unequal precision to eliminate the error in the distance measurement. Further, the effect of atmospheric delay on distance measurements is considered, leading to further correction of the inversion results. At last, an airborne experiment was carried out in a sea area near Qinhuangdao, China on 14 March 2019. The results showed that the ranging accuracy can reach 0.9066 m, which achieved an excellent ranging accuracy on 1.57-μm IPDA Lidar and met the requirement for high-precision CO_2_ column length inversion.

## 1. Introduction

One of the biggest problems for mankind is the greenhouse effect. Since the Industrial Revolution in the last century, the global average CO_2_ concentration has risen from 278 ppm before industry to 410 ppm in 2020. Specifically, in April 2014, the monthly CO_2_ concentration in the atmosphere of the Northern Hemisphere exceeded 400 ppm for the first time [1]. The increased concentration of CO_2_ gas is mainly due to man-made factors, such as the burning of fossil fuels and changes in land use. Furthermore, half of all CO_2_ produced by human activities is absorbed by the oceans and land. To better understand the movement of carbon among the atmosphere, land, and oceans, including carbon sources and sinks, precise measurements of the tropospheric CO_2_ mix ratio are needed. The measurement accuracy of atmospheric CO_2_ column concentration [2,3,4] needs to reach ~0.3% to improve the accuracy of carbon source and sink calculation. This goal is difficult to achieve for passive spectrometers using reflected sunlight, due to scattering from atmospheric aerosols and low coverage at high latitudes [5,6,7,8,9,10,11,12,13,14,15,16].

To meet these requirements, integrated path differential absorption (IPDA) Lidar technology was proposed to measure the CO_2_ column concentration in the laser path. The strong backscatter of a hard target and the received strong signals were used to minimize the Lidar power and achieve high signal-to-noise ratios. For the inversion of the CO_2_ column concentration, an optical inversion model needs to be constructed. The determination of the upper and lower bounds of the denominator integral in the model involves three parts: aircraft inertial navigation information, ranging information, and meteorological parameters. Among the three parameters, the ranging information plays a very important role in the entire model. When the ranging error is less than 3 m, the error of the estimation of the CO_2_ column concentration using the IPDA Lidar system in space [17] is less than 1%. Therefore, improving the CO_2_ column concentration inversion from the perspective of the accurate measurement of the CO_2_ column length is feasible.

At present, there are limited studies on high-precision distance inversion in CO_2_ column concentration measurements with the IPDA Lidar system. In 2011, Amediek et al. [18] experimented with a 1.57-μm band type and achieved a ranging accuracy of 2.8 m. In addition, Refaat et al. [19,20] carried out a ranging experiment with a 2-μm band type in 2014 and successfully obtained a ranging accuracy within 3 m, which met the requirements for CO_2_ column concentration inversion. Other researchers have conducted extensive studies on pure-airborne or satellite-borne altimetry. Regarding space-borne altimetry, Ma Yue et al. [21] conducted system data processing and error analysis of the Icesat-1 satellite and measured the height of the ice sheet in the Greenland Island area with high precision. Wei Yao et al. [22] completed tree species classification and estimation by using airborne full waveform Lidar data. Carabajal et al. [23] studied the waveform decomposition, geometric coordinate solution, and elevation accuracy verification of shuttle laser altimeter Lidar altimetry data. Herzfeld et al. [24] studied the processing of micro pulse photon-counting Lidar altimetry data of the launched Icesat-2.

Among these extensive studies, the central issue is the measurement of the time center of mass of the Lidar pulse waveform [25,26,27,28,29,30]. A common Lidar pulse ranging method is the early-late gate or split-gate method [31], which is dependent on waveform symmetry. In this method, a histogram composed of the photons received by the pulse is divided into two parts, namely, early and late gates. Then, through repeated iterations, the number of photons in the respective gate becomes equal. Another theory involves conducting least-square fitting between the waveform of the returned signal and the expected one. This method is simple, but the shape and the transit time of the waveform of the returned signal need to be known in advance. The above methods are subject to the asymmetry of echo pulse signals and lack of the information on the shape of such signal. More importantly, previous scholars have had limited information caused by the sampling mechanism of the platform, with a single observation in the same place. However, the IPDA system can simultaneously detect two consecutive laser beams at a noticeably short interval on a similar observed surface. Therefore, an error cancellation mechanism is needed for two-band data measurements from a single observation, which has not been carried out by previous scholars.

From the above research and combined with the actual conditions of our study area, this study proposes a practical method to improve the accuracy of the distance measurement. The main advantages of this method are as follows: First, from the fusion of processing methods in different data stages (including waveform function autocorrelation detection, adaptive Gaussian filtering, generalized Gaussian decomposition, and Levenberg–Marquardt (LM) fitting based on the generalized Gaussian distribution), the time parameters of echo pulses are estimated accurately. Second, from the redundant observation of the two-band pulse signal, the error elimination theory is introduced to eliminate the gross error in the distance measurement length of the column and further modify range values using atmospheric delay correction. Our goal is to accurately measure the distance of the column length and to obtain a distance measurement than can meet the requirement for high-precision CO_2_ concentration. In addition, the proposed method is a complete idea of data processing and verification that serves as a reference for subsequent studies of the same type. Excellent column length results have been quantitatively obtained with 1.57-μm band IPDA Lidar by adopting the theory in subsequent experiments.

## 2. Methodology

### 2.1. Principle of the IPDA

Figure 1 shows the detection mechanism of the airborne IPDA Lidar for detecting CO_2_, which emits two Lidar pulses at 200 μs interval to form an observation pair, namely, on-line and off-line wavelength pulse signals. During the propagation of the laser beam in the atmosphere, the on-line and off-line wavelength pulses are attenuated by extinction. In addition, the on-line wavelength laser is sensitive to the detected CO_2_ gas in the atmosphere, thus producing energy attenuation of the returned signal, which is different from the latter one. Therefore, the on-line and off-line wavelength signal data can be combined to conduct high-precision inversion of the CO_2_ column concentration by the following equation.
(1)$$XCO_{2}=\frac{DAOD}{IWF}=\frac{\frac{1}{2}\cdot ln\left(\frac{P_{Off}\;.\;E_{On}}{P_{On\;}\;.\;\;E_{Off}}\right)\;}{\int_{p=0}^{P_{SFC}}WF_{CO_{2}}\left(p\right)\cdot dp},$$
where $XCO_{2}$ is the concentration of CO_2_ after inversion; *DAOD* is differential absorption optical thickness; *IWF* is the integral weight function; *P* and *E* are the ground echo and output energies, respectively, which are measured at a corresponding on-line or off-line wavelength; $WF_{CO_{2}}\left(p\right)$ is the weighted function of CO_2_; and $P_{SFC}$ and P=0 are the integral upper and lower limit parameters determined by ranging information, meteorological parameters, and aircraft inertial guidance information.

Equation (1) shows that the accuracy of the upper and lower limits of the integral is closely related to the ranging information. Moreover, the improvement of the ranging information accuracy can affect the inversion accuracy of the CO_2_ column concentration; that is, the ranging accuracy of IPDA needs to be 3 m or less [19,20].

In the process of detecting a single CO_2_ column concentration, the pulsed laser emitted by the IPDA Lidar needs to go through atmospheric transmission, target reflection, echo signal reception, photoelectric conversion, and other processes successively. Consequently, the signals are mixed with a substantial amount of noise. The aircraft’s attitude roll and course change during flight may also lead to the loss of signals. Therefore, the data need to be preprocessed through three steps: Signal validity detection, Noise estimation, and Smoothing filtering. It is completed by Gaussian decomposition and LM fitting based on the generalized Gaussian distribution. Four steps are as followed: namely, dual-band measurement error correction, atmospheric delay correction, Lidar pointing angle correction, and aircraft attitude angle correction, are used. The objectives are to complete the high-precision ranging and data comparison verification of the data in the research area and to achieve a high-precision inversion of the CO_2_ column concentration. Figure 2 shows the data processing flow chart.

### 2.2. Validity Detection of the Lidar Pulse-Echo Signal

In the study area on the sea level, the laser emitted by the Lidar transmitter is affected by the aircraft’s attitude angle. Thus, the Lidar beam forms an incident angle with the sea level. When the comprehensive influence of the roll and pitch angle reaches a certain threshold, the incident angle of the emitting Lidar beam is extremely large, thereby the emitted signal could not be captured by the receiver. This comprehensive influence can be calculated by using Equation (2), which represents the angle between the actual laser beam direction and the perpendicular direction. In addition, the echo energy of some signals are lower than that of the detector threshold due to the absorption effect of the sea water, thereby resulting in signal loss, and the fluctuation of airborne elevation in a short time will also cause the decline of signal quality. Therefore, for the case of fluctuation in data validity, we designed the detection mechanism of echo signal validity in the following paragraph and avoided data that could not meet the requirement of inversion of the CO_2_ column concentration.
(2)Lpa = arcos(cos(Pa)∗cos(Ra)),
where Lpa is the angle between the actual laser beam direction and the perpendicular direction; Pa is the pitch angle; and Ra is the roll angle.

The reference waveforms of the 1.57-μm on-line and off-line band beams are detected in correlation with their corresponding echo signals. The waveform data of a 10% proportion of energy are recorded at the moment of laser emission, which is mainly used to calculate the waveform energy of the current pulse pair so that it can be chosen as the reference. Thus, the intensity range of the Lidar echo signal from sea level reflections was identified. When the difference of the detected time center for the double beams is less than a certain threshold, the observed signal is confirmed to be effective, that is, the CO_2_ column concentration can be reversely determined. Hence, the data of the abnormal observation pair need to be removed. The corresponding mathematical expression is as follows:(3)$$R_{f\left(t_{i}\right)}={\left|\left|{\left(f\left(\tau \right)-f\left(t_{i}\right)\right)}^{2}\right|\right|}_{1},$$
where f(τ) is a one-dimensional sequence from reference waveforms, $f\left(t_{i}\right)$ is a one-dimensional sequence of the same scale at different times, and i=1,2…,n.

Finally, the time reference center of the effective echo signal is the point corresponding to the minimum value in the one-dimensional group.

### 2.3. Lidar Pulse-Echo Noise Estimation and Data Filtering

In general, the pulse-echo signal received by the system detector can be regarded as the superposition of multiple Gaussian functions and noise signals. Given the presence of clouds in the atmosphere, the signal not only generates multiple echoes but also distorts the signal. Therefore, the echo signal can be expressed as:(4)$$W\left(t\right)=\varepsilon +\sum_{m=1}^{N_{p}}A_{m}e^{-\;\frac{{\left(t-t_{m}\right)}^{2}}{2\sigma_{m}^{2}}},$$

For the data filtering of the echo signal, the noise signal that is effectively aliased on the received pulse-echo signal can be regarded as Gaussian white noise. The weighted moving average method is used to realize the smooth filtering of the noise signal. Considering that the received pulse-echo signal is the sum of a series of Gaussian functions, the normalized Gaussian function is selected as the weight function of smooth filtering. The corresponding mathematical expression is as follows:(5)$$g\left(t\right)=\frac{1}{\sqrt{2\pi }\sigma }e^{\left(-\frac{t^{2}}{2\sigma^{2}}\right)},$$
where σ is the root mean square pulse width of the Gaussian filter function. The Gaussian filtering process is equivalent to convolving the echo function *w(t)* with the filtering function g(t). The formula expression of the filtered signal is as follows:(6)$$W\left(t\right)*g\left(t\right)=\varepsilon \left(t\right)*g\left(t\right)+\sum_{m=1}^{N_{p}}\frac{A_{m}\sigma_{m}}{\sqrt{\sigma^{2}+\sigma_{m}^{2}}}e^{-\;\frac{{\left(t-t_{m}\right)}^{2}}{2(\sigma^{2}+\sigma_{m}{}^{2})}},$$

To further remove noise and apply adaptive filtering, the root mean-square pulse width of the adaptive filter is used [22] as follows:(7)$$\sigma =\left[k\times \frac{\sigma_{n}}{\sigma_{s}}\right],\;$$
where k is a constant positive integer and usually we choose 2~3 times the pulse width of the reference waveforms. The [ ] symbol is expressed as an integral function. $\sigma_{n}$ is the variance estimation result of the background noise, and $\sigma_{s}$ is the standard deviation value of the current window data.

### 2.4. Gaussian Decomposition and Fitting Based on Generalized Gaussian Distribution

From the above theory, Gaussian decomposition is used to fit the superimposed data of multiple waveforms into superimposed curves of multiple Gaussian functions. Thus, the Gaussian parameters of each waveform are obtained, and the parameterization of each waveform is completed, which provides a basis for subsequent research. The parameters obtained by Gaussian decomposition are generally regarded as initial estimation parameters. To further obtain accurate Gaussian parameters, we provide the initial estimate of the Gaussian component in the LM algorithm based on the generalized Gaussian distribution and apply the corresponding constraint conditions to obtain the non-linear fitting results that meet the accuracy requirements.

We use Gaussian decomposition theory to decompose the waveform and assume that the reflected pulse is bell-shaped. Then, the inflection point of the filtered waveform data and the position of the initial wave peak can be obtained by solving the first and second partial derivatives of the distribution function and setting them equal to 0. The first and second partial derivatives are as follows:(8)$$\frac{\partial W_{m}}{\partial t}=-A_{m}e^{-\;\frac{{\left(t-t_{m}\right)}^{2}}{2\sigma_{m}^{2}}}\times \;\frac{{\left(t-t_{m}\right)}^{2}}{\sigma_{m}^{2}},\;$$
(9)$$\frac{\;\partial^{2}W_{m}}{\partial t^{2}}=A_{m}e^{-\;\frac{{\left(t-t_{m}\right)}^{2}}{2\sigma_{m}^{2}}}\times \;\left[\frac{{\left(t-t_{m}\right)}^{2}}{\sigma_{m}^{2}}-\frac{1}{\sigma_{m}^{2}}\right],\;$$
(10)$$W_{GGm}=A_{m}e^{-\;\frac{{\left(t-t_{m}\right)}^{\left(\alpha_{n}\right)2}}{2\sigma_{m}^{2}}},$$
where $A_{m}$ is the mth Gaussian component amplitude, $\frac{\partial W_{m}}{\partial t}$ and $\frac{\;\partial^{2}W_{m}}{\partial t^{2}}$ are the mth Gaussian components of the first and second partial derivatives, $W_{GGm}$ is a generalized Gaussian expression for the mth component, and $\alpha_{n}$ is the shape parameter, in which $\alpha_{n}=1.00,\;1.02,\;1.04\ldots 1.98,\;2.00.$

To determine the initial wave peak Gaussian parameters, the findpeaks function from MATLAB is applied. On this basis, the information of a single Gaussian crest is determined by combining the information of the inflection point. By applying the corresponding constraints, according to the LM non-linear fit based on the generalized Gaussian distribution, we perform a more refined waveform fit for each initial estimate based on the above Gaussian decomposition. With reference to generalized Gaussian theory, we introduce the shape parameter α to change the shape of the current iteration of the Gaussian component waveform, making it similar to the real data waveform.

### 2.5. Double Measurement Adjustment Ranging Optimization

The IPDA Lidar transmitter emits 1.57-μm on-line and off-line beams in parallel successively at a 200 μs interval. Thus, this transmitter can be regarded as two consecutive distance measurements of the same target, thereby generating redundant observations of the ranging information. Furthermore, although the LM algorithm based on the generalized Gaussian method is optimized; the distance measurement observation value still has error relative to the true value. Since the distances were measured separately for on-line and off-line bands, the distance measurement values generated by the two observations must be contradictory. To balance this contradiction, we introduce the direct adjustment theory of unequal precision.

Regarding measurement adjustment, we generally consider weight as the relative evaluation index of the reliability of observation results, which is inversely proportional to the square of the median error. The most probable value formula is used to obtain the high-precision range values. The weight and the most probable value formula are as follows:(11)$$P_{n}=\frac{1}{m_{n}^{2}}$$
(12)$$x=\frac{\sum_{n=1}^{2}P_{n}\;\times \;L}{\sum_{n=1}^{2}P_{n}},$$

In Equation (11) $m_{n}^{2}$ is the median error corresponding to the on-line or off- line bands; $P_{n}$ is the weight corresponding to the on-line or off- line bands; *L* is the measurement value of the on-line or off-line bands processed from the above data, and x is the optimized high-precision ranging value.

The laser beam crosses the atmosphere and reaches the surface, thereby accumulating atmospherically induced errors at different heights. Thus, we selected 100 sampling points (approximately 120-m spatial distance) before the laser and surface interaction at the screen and iterated the 10 sampling point window data to obtain small standard deviation values to represent the weight of the measurement tool accuracy of the 1.57-μm on-line and off-line laser beam.

### 2.6. Atmospheric Delay Correction

The observational accuracy of satellite laser ranging technology is mainly affected by the propagation model error of the signal as it passes through the troposphere and stratosphere, that is, the refractive delay of the neutral atmosphere. Additionally, the data of our airborne IPDA Lidar detection at the altitude of 6.9 km in the test area are influenced by this factor, so we introduce the refractive index model currently recognized internationally as Ciddor [32] and the ray tracing method to calculate the atmospheric refractive error of the airborne IPDA Lidar range obtained using the Ciddor atmospheric refraction model.
(13)$$d_{atm}^{z}=10^{-6}\times \int_{r_{s}}^{r_{a}}Ndz,$$

Equation (13) is the schematic formula for the refractive delay of the neutral atmosphere of the zenith direction electromagnetic waves, where *N* is the group refractive index difference, $r_{s}$ is the station’s geocentric moment, ra is the aircraft GPS altitude, and $d_{atm}^{z}$ is the path integral along the zenith direction.

According to the International Association of Geodesy (IAG) and International Union of Geodesy and Geophysics (IUGG) 1999 resolution, the theoretical representation of the difference in the atmospheric mass refractive index in the optical and Near-Infrared bands should be in the form of Ciddor [32], namely,
(14)$$N=\;\frac{\rho_{a}}{\rho_{axs}}\;\left(n_{axs}-1\right)+\frac{\rho_{w}}{\rho_{ws}}\;\left(n_{ws}-1\right),$$
(15)$$n_{axs}-1=10^{-2}\times \left[\frac{k_{1}\left(k_{0}+\sigma^{2}\right)}{{\left(k_{0}-\sigma^{2}\right)}^{2}}+\frac{k_{3}\left(k_{2}+\sigma^{2}\right)}{{\left(k_{2}-\sigma^{2}\right)}^{2}}\right]\times \left[1+0.534\times 10^{-6}\left(x_{c}-450\right)\right],$$
(16)$$n_{ws}-1=10^{-2}\times cf\left(w_{0}+3w_{1}\sigma^{2}+5w_{2}\sigma^{4}+7w_{3}\sigma^{6}\right),$$

In Equation (14) $\rho_{axs}$ is the density of dry air at 15 °C, 101325 Pa, and $\chi_{w}=0$, and $\rho_{ws}$ is the density of pure water vapor at 20 °C and $\chi_{w}=1$; $\rho_{a}$ and $\rho_{w}$ are the densities of the dry air component and the water vapor component of moist air, derived from Equation (17) for the actual conditions; $n_{axs}-1$ is the difference in the refractive index of the water gas population under standard atmospheric conditions; σ is the wavenumber in inverse micrometers, and $x_{c}$ is the CO_2_ concentration; $n_{ws}-1$ is the refractive index difference of the dry-atmosphere population at standard atmospheric conditions, $\;k_{0}\;,\;k_{1},\;\;k_{2},\;\;k_{3}\;,\;\;w_{0}\;,\;cf\;,w_{1}\;,$
$w_{2}\;$, and $w_{3}$ are constants and are summarized in Table 1.
(17)$$\rho =\left(\frac{{ΡM}_{a}}{ZRT}\right)\left[1-\chi_{w}\left(1-\frac{M_{w}}{M_{a}}\right)\right]$$
(18)$$M_{a}=10^{-3}\times \left[28.9635+12.011\times 10^{-6}\left(x_{c}-400\right)\right]$$
(19)$$Z=1-\left(\frac{P}{T}\right)\left[a_{0}+a_{1}t+a_{2}t^{2}+\left(b_{0}+b_{1}t\right)\chi_{w}+\left(c_{0}+c_{1}t\right)\chi_{w}^{2}\right]$$
(20)$$svp=\exp\left(AT^{2}+BT+C+D/T\right)$$
(21)$$f=\alpha +\beta Ρ+\gamma t^{2}$$
(22)$$\chi_{w}=\left(f\;h\;svp\right)/P,$$

In Equation (17), *Ρ* is the total pressure in pascals, R is the gas constant, T is the temperature in degrees Kelvin, and $M_{w}$ is the molar mass of water vapor. In Equation (19), *Z* is the compressibility of moist air, and t is the temperature in degree Celsius. In Equation (20), svp is the saturation vapor pressure of water vapor in air at temperature T. In Equation. (21), f is the enhancement factor of water vapor in air. In Equation (22), *h* is the fractional humidity (between 0 and 1). $a_{0},a_{1},\;\;a_{2},\;\;b_{0},b_{1},\;\;c_{0},c_{1},\;\;d,\;e,\;A,\;B,\;C\;,\;D,R,\;\;M_{w},\;\alpha ,\;\beta$ and γ are constants from their respective formulas, which are summarized in Table 1.

Therefore, we combine the available atmospheric detection data and calculate the atmospheric delay values for different altitude layers by using Equation (13) through Equation (22).

## 3. Experimental Area and Data

On 14 March 2019, an aircraft carrying IPDA Lidar equipment was used to detect the CO_2_ column concentration according to a preset route. The overall flight trajectory of the aircraft mainly covered part of the sea area around Qinhuangdao in China (as shown in Figure 3), with the latitude at [39.66178894 N, 40.02510834 N] and the longitude at [119.447319 E, 120.1018372 E]. We calculated the statistics for the changes in the overall altitude of the aircraft in the research area by employing the aircraft’s GPS elevation data. During the 853-s flight, the maximum, minimum, and average altitudes of the aircraft were 6814, 6785, and 6799.5 m, respectively. Therefore, the aircraft had a smooth flight on the flight trajectory. Furthermore, the data recorded by the on-board equipment, i.e., temperature, pressure, and water vapor content, during the test period can be considered consistent, which were 14.99 °C, 1013.3 hPa, and 1.8321 g/kg.

The experimental data obtained in the research area comprise six files, in which a single file contains 4000 data packets. One packet contains an observation pair, and each packet contains 22,000 float data points, namely, on-line and off-line wavelength pulse-echo signals, corresponding to 11,000 float data points. The first 11 data of each observation pair also contain the current aircraft attitude information, GPS cycle second information, longitude and latitude, altitude, course speed, and other information.

Figure 4 shows the original echo data of an offshore observation pair in the overall data. The single observation pair contains the on-line and off-line echo data. Figure 4a is mainly used to show the Lidar emission pulse energy of reference waveforms, as Figure 4b shows that the energy attenuation after the interaction between the online pulse and CO_2_ in the atmosphere is extremely high. Table 2 shows some hardware parameters of the IPDA Lidar [33]. The pitch, roll and laser pointing angle information of the aircraft at different times in the study area are shown in Figure 5.

## 4. Experimental Results

### 4.1. Validation Data

The measured height of the CO_2_ column concentration is based on the local mean sea level from the WGS84 elevation reference. According to geodetic theory, the elevation data of the local mean sea level are given by the adjacent tide station-the Qinhuangdao tide station. Moreover, the average sea level changes in elevation by only a few centimeters over a distance of 100 km from the tidal station, so it can be used as the mean sea level height of the research area.

The elevation datum of the tide survey station in China is the 1985 elevation datum, and the obtained mean sea level is the normal height with the quasi-geoid as the starting surface. The altitude of the aircraft GPS is the WGS84 elevation reference, and the obtained altitude is the earth elevation starting from the WGS84 ellipsoid. Thus, the problem is that the elevation starting datum is different. Therefore, the EGM2008 gravity model is used to obtain the elevation anomalies of each observation point [34], and then, the average sea level height of each point under the WGS84 elevation datum is obtained through combination with the data of the tide station to unify the elevation starting datum and verify it with the subsequent data.

During the flight of an aircraft, the Lidar beam is not always perpendicular to the sea level due to the existence of roll and pitch angles, so offset occurs because of the attitude angle. Figure 6 shows the geometric model of this offset. The offset value is an extra distance between the distance measurement value of the Lidar column and the plane perpendicular length. Compared with the 3 m ranging accuracy requirement for 1 ppm CO_2_ concentration inversion, the GPS elevation accuracy of 0.2 m is very high [35]. Therefore, we need to combine the geometric model to convert the distance of CO_2_ column into vertical distance, and then we can verify the accuracy of CO_2_ column length by comparing with GPS elevation.

### 4.2. Data Processing Results

From the above theory, the single waveform data in the research area are processed, and the results are shown in Figure 7. Through the estimation of the overall data, the two threshold background noise lines of the off- and on-line bands are extracted to represent the interval threshold of the effective waveform data. After the original data presented in Figure 7 are processed following the method, the fitting data after data processing are obtained. In the data interval of the effective echo waveform, the time centers of the gravity parameters after fitting the off- and on-line bands are 5667.446 and 5666.859, respectively. Combining the off-line and on-line time centers of gravity, the time center of gravity parameter is 5667.218 according to the introduced elimination error theory.

Other data, such as wavelength, CO_2_ concentration, temperature, pressure, and humidity, are used simultaneously in calculating the error caused by atmospheric delay. However, factors other than the CO_2_ concentration and altitude were identified.

Hence, we analyzed the sum of atmospheric delay effects from surface to a specific altitude of 6.8 km with different CO_2_ concentration, based on the off-line band. However, the result in Figure 8a shows that this effect influence by different CO_2_ concentration is on the order of magnitude of 10^−6^ m, which is very small. Then, we calculate the atmospheric delay for different altitudes and choose 420 ppm as the CO_2_ concentration. Figure 8b shows the distribution of atmospheric delay of off-line band (1572.085 nm) from the near-Earth surface. It produces an effect of 1.355 m under the altitude of 6.8 km, occupying half of the whole atmospheric delay value [36].

From a comparison of the two types of data in Figure 9a, the overall data in the study area are divided into three parts: valid, invalid, and cloud data. The double-band returned pulse signal can be used to invert the CO_2_ column concentration data known as the position valid data. The invalid data occur mainly because the comprehensive influence of the extremely large attitude angle and the lower echo signal. Hence, some of the data cannot be used to invert the CO_2_ column concentration. Considering that cloud vapor floating on the sea is extremely condensed, the cloud thickness exceeding a certain threshold disables the Lidar beam to reach the sea level, and cloud echo data are generated.

Figure 9b focuses on the local details of the valid data in Figure 9a, and the red and black dots have the same meaning as those in Figure 9a. Figure 9b shows that the CO_2_ column length is far beyond the height recorded by the airborne GPS on both sides of the aircraft’s steering area. As mentioned above, this limitation is caused by the combined influence of the attitude angle when the aircraft turns. However, the CO_2_ column length value that is corrected for consistency is extremely close to the airborne GPS elevation, indicating that our measurement of the CO_2_ column length is effective even in the aircraft’s steering area.

To further reveal the relationship between the CO_2_ column length corrected at each point and the height of the airborne GPS, we provide Figure 9c. According to the statistics, during the 795-s flight in the research area, 15,918 valid data points were collected. At the corresponding time, the fluctuation range of the GPS height value and corrected CO_2_ altimetry data difference is located at [−5.596, 7.372], in which 0.4% of the data set is located at [−5.596, −3], 99.50% of the data set is located at [−3, 3], and 0.1% of the data set is located at [3, 7.372]. In addition, the study area is a nearly calm sea; hence, the error of the ranging accuracy can be expressed as the data standard deviation of the difference between the GPS height value and the corrected CO_2_ altimeter data. That is, the overall standard deviation of the data difference in Figure 9c is 0.9066 m. In Figure 9c, the distribution of the most difference value is stable at [−3, 3], except for the part marked by the green rectangle. At the position of green rectangle box 1, the elevation fluctuates for a short time and the pointing angle of Lidar is relatively stable. Therefore, this verifies that the jitter of elevation in a short time will affect the quality of data. Subsequently, in Figure 9c, the accuracy of green rectangle box 3, 4, 5, 6, and 7 located at the turning point of the aircraft (assuming that the rectangle box counts from left to right) are significantly reduced. At these positions, a common situation exists, namely, the jitter of aircraft elevation in Figure 9f and the jitter of aircraft Lidar pointing angle in Figure 9e. At green rectangle box 2, the plane is also in the state of turning. However, at this time, the short-term fluctuation of elevation in Figure 9f is relatively mild, and the accuracy of Figure 9c does not show an obvious upward and downward trend. Therefore, it can be concluded that the combination of the fluctuation of elevation and Lidar pointing angle in a short time will aggravate the degradation of inversion quality. Individually, the fluctuation of elevation appears to have a greater effect than that of pointing angle.

For the cloud data, we conducted a statistical analysis on a single cloud surface and depicted the resulting data in Figure 10. This cloud has a total of 103 valid data points, and the overall standard deviation is 49.314 m. This result is not surprising mainly because the upper surface of the cloud is a random bump rather than a plane, and the thickness of the cloud at different locations has different effects on Lidar penetration.

### 4.3. Error Analysis

The overall error in the flight measurement is mainly divided into two parts: range and data validation errors. The Lidar column measurement error mainly has the following factors: limited detection ability of the detector, and error caused by the signal sampling rate. For the data validation error, the main error source is the uncertainty of the altitude value of GPS. During the flight of the aircraft, GPS dynamic positioning causes a certain amount of error in the recorded altitude, and the error value is 10–20 cm. The second part is the change in the attitude angle of the aircraft, which causes the column length to deviate from the vertical direction. In the data validation, we convert the CO_2_ column length to the vertical direction according to the geometric model, and then compare with the GPS elevation. The height of GPS is located on the normal line with respect to the sea level, but we convert the distance to the perpendicular line, so this error exists, which is also known as the vertical deviation of altimetry. Fortunately, some scholars [37] have found that most of the impact is at the centimeter level, which is small compared with that in our requirements for distance measurement accuracy, so this error was ignored.

## 5. Conclusions

For an IPDA Lidar system, one of the preconditions for measuring CO_2_ column concentration is to understand the Lidar column length from the aircraft to its spot scattering surface. To improve the precision of CO_2_ column concentration inversion from the perspective of height measurement, a data flow processing method is proposed. In the early data preprocessing process stages, such as signal detection, noise estimation, and waveform smoothing, the method restores the true waveform information to the maximum extent. The theory includes Gaussian waveform decomposition, waveform parameter estimation, and LM fitting based on the generalized Gaussian distribution, in addition to the echo signal digitization and high precision of approximate distance center. However, even after single-pulse optimization, the gravity center value of the high-precision distance measurement time has error compared with the theoretical true value. Therefore, in the later stage, we combine the optimized ranging value of on-line and off-line pulse-echo signals and the weight extracted from the original data and corrected atmospheric delay values to further eliminate the error compared with the truth value. Furthermore, the distance accuracy of the 6.9-km elevation data set is 0.9066 m. Furthermore, at the corresponding time, the fluctuation range of the difference between the GPS height value and corrected CO_2_ altimetry data is located at [−5.596, 7.372], in which 99.50% of the data set is located at [−3, 3]. In addition, we also analyzed the cloud data in the experimental area, due to the random upper surface and the different thickness of the cloud, the overall standard deviation is larger than that of the sea, up to 49.314 m. The proposed method not only meets the requirement of high-precision CO_2_ altimetry but also improves the precision of the distance measurement compared with similar work done before.

In the aspect of improving the accuracy of the distance measurement and then improving the concentration of high-precision carbon dioxide columns onboard the inversion, the accuracy of the current algorithm verification is sufficient to meet the demand without considering the atmospheric factor. Second, for data validation errors, we will further correct the elevation value of GPS dynamic positioning and the deviation of the vertical line of height measurement. In the future, we believe that the distance measurement accuracy of the column length will be greatly improved through error correction from these two perspectives, which will also provide strong support for the high-precision inversion of CO_2_ column concentration.

In addition, we will also use the IPDA Lidar data to analyze clouds on the basis of actual conditions, such as cloud height and thickness. In this way, we can fully realize the potential of the IPDA Lidar system and obtain diverse results to support CO_2_ inversion. In summary, the results show that the set of algorithms can achieve an excellent ranging accuracy, which is sufficient for columnar CO_2_ density estimation, without the need for separate ranging channels and associated additional hardware.

## Figures and Tables

**Figure 1 sensors-20-05887-f001:**
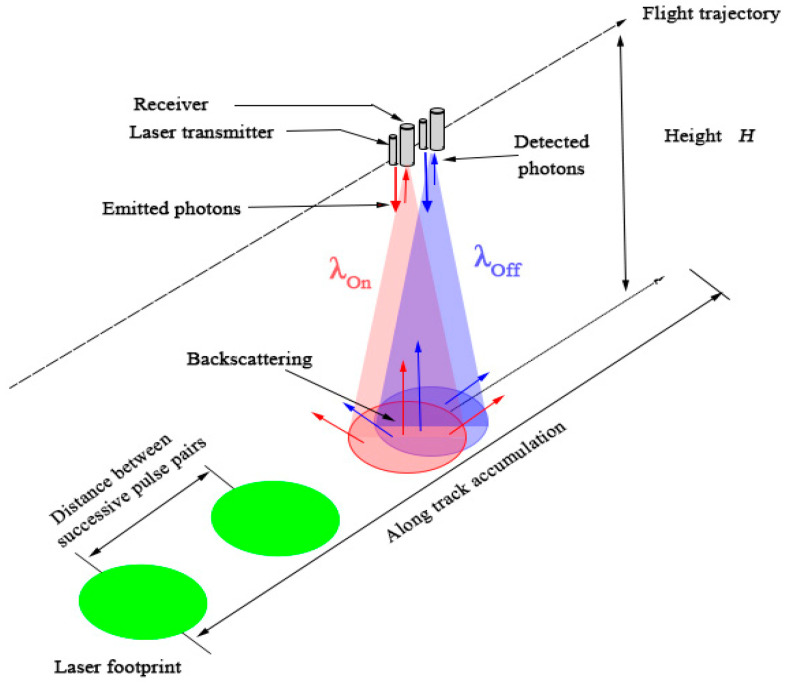
Detection mechanism of the airborne integrated path differential absorption (IPDA) Lidar for detecting CO_2_.

**Figure 2 sensors-20-05887-f002:**
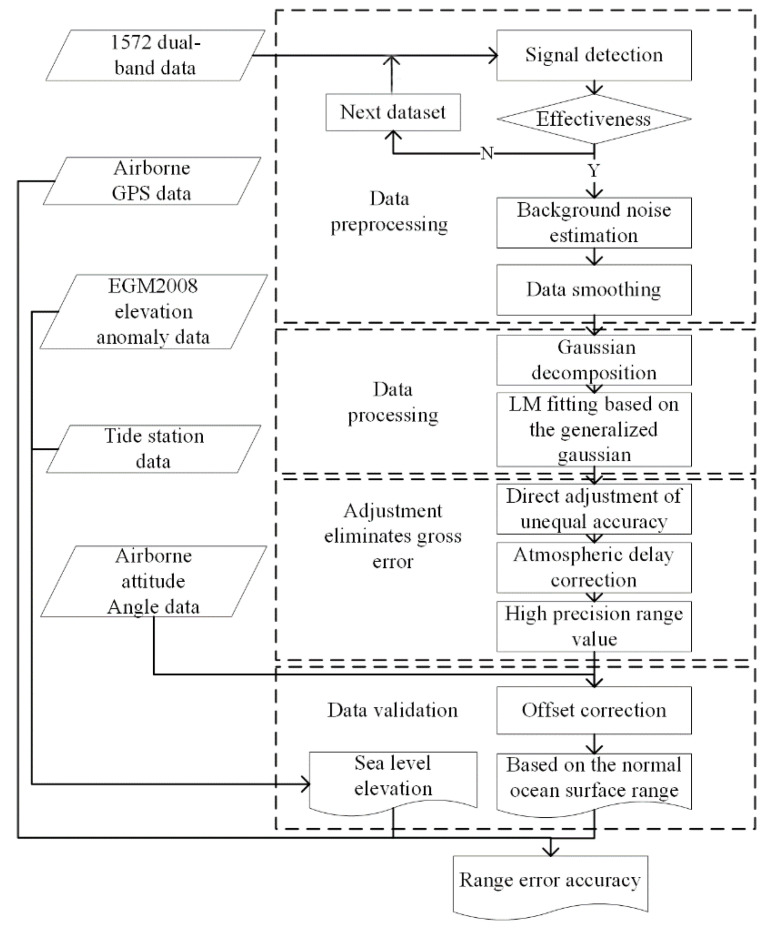
Flowchart of data processing.

**Figure 3 sensors-20-05887-f003:**
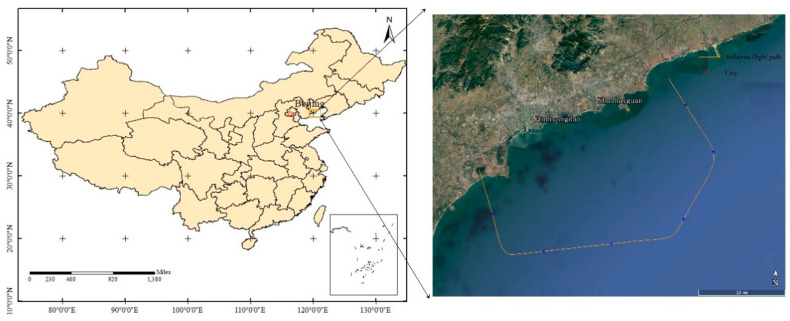
Flight trajectory of aircraft in the study area. The aircraft moves along the direction of the arrow on the path to collect data successively.

**Figure 4 sensors-20-05887-f004:**
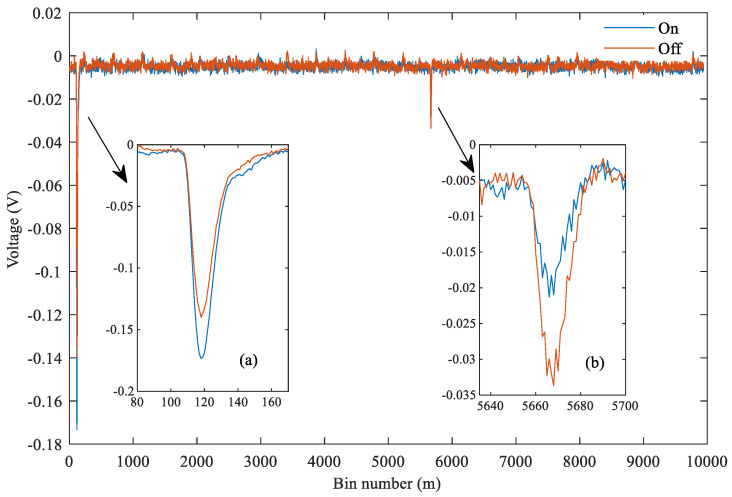
Superimposed on-line and off-line raw signal data. (**a**) Signal waveform shape of the specified proportion saved during the Lidar pulse emission. (**b**) On-line and off-line sea level echo signals.

**Figure 5 sensors-20-05887-f005:**
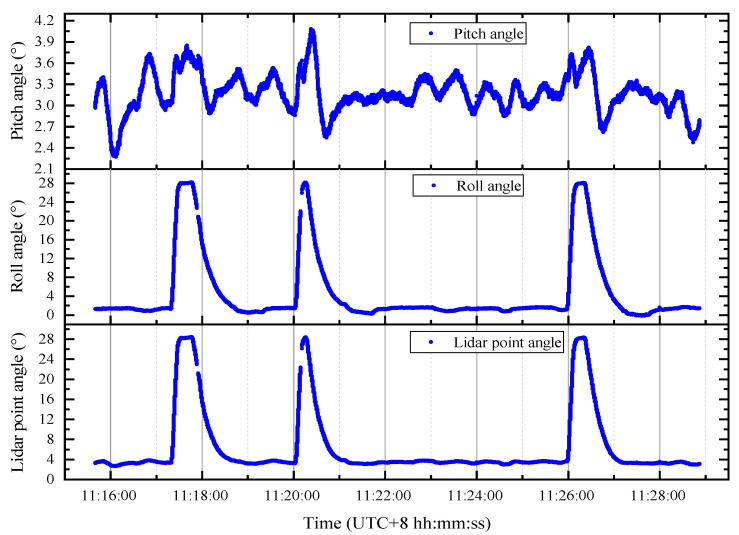
Airborne angle information. The corrected pitch angle, roll angle, and laser pointing angle are shown, respectively. The angle between the laser beam pointing and the vertical line is called the laser pointing angle, which is calculated from the pitch and the roll angle by using Equation (2). There are two breakpoints in the graph because there is one second missing data at two different locations in the original data.

**Figure 6 sensors-20-05887-f006:**
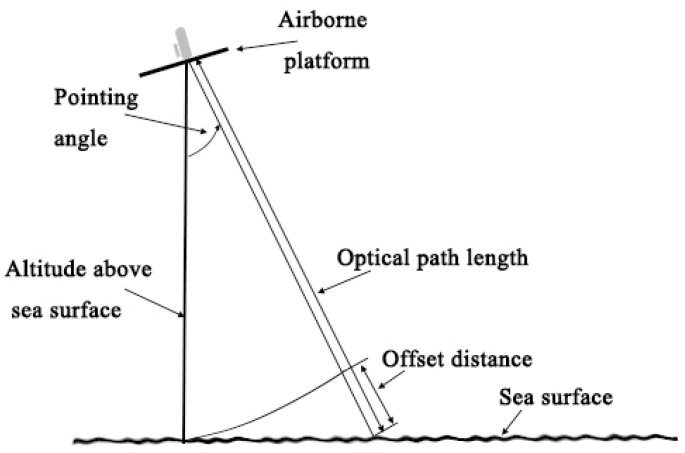
A geographic model of distance offset caused by pointing angles. The point angle represents the combined effect of the pitch and roll angles. The altitude above the sea level is the distance perpendicular to the sea level from the plane.

**Figure 7 sensors-20-05887-f007:**
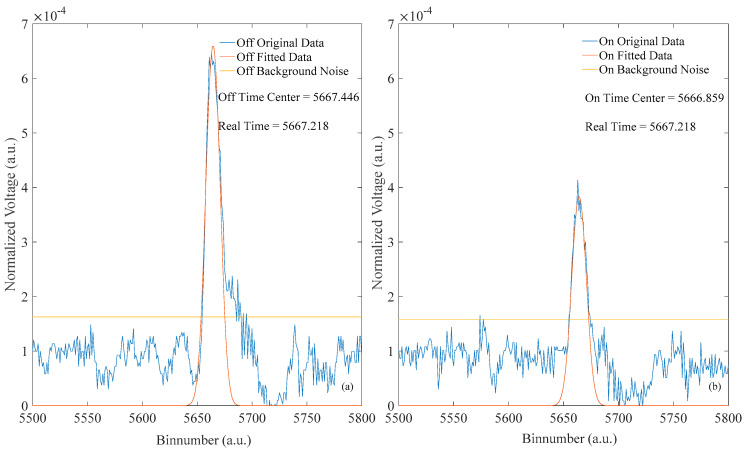
Waveform data processing results. (**a**,**b**) show the results of data processing in off-line and on-line, respectively. The off-line (on-line) time center represents the value after data processing and parameter optimization in the (**a**) or (**b**). Real-time represents the value after balancing the contradictory values of the dual-band time-center of gravity by introducing the error elimination theory.

**Figure 8 sensors-20-05887-f008:**
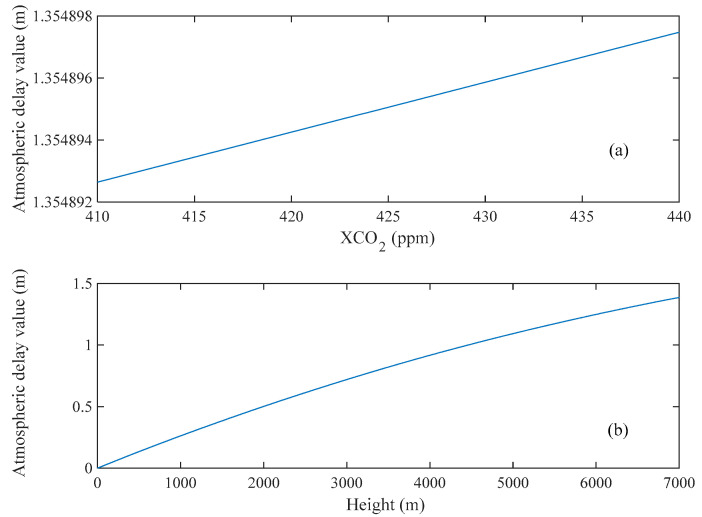
Results of the calculation of the atmospheric delay correction term. (**a**) shows the atmospheric delay results for different CO_2_ concentrations at the altitude of 6.8 km. (**b**) shows the atmospheric delay results for different altitudes calculated with a CO_2_ concentration of 420 ppm.

**Figure 9 sensors-20-05887-f009:**
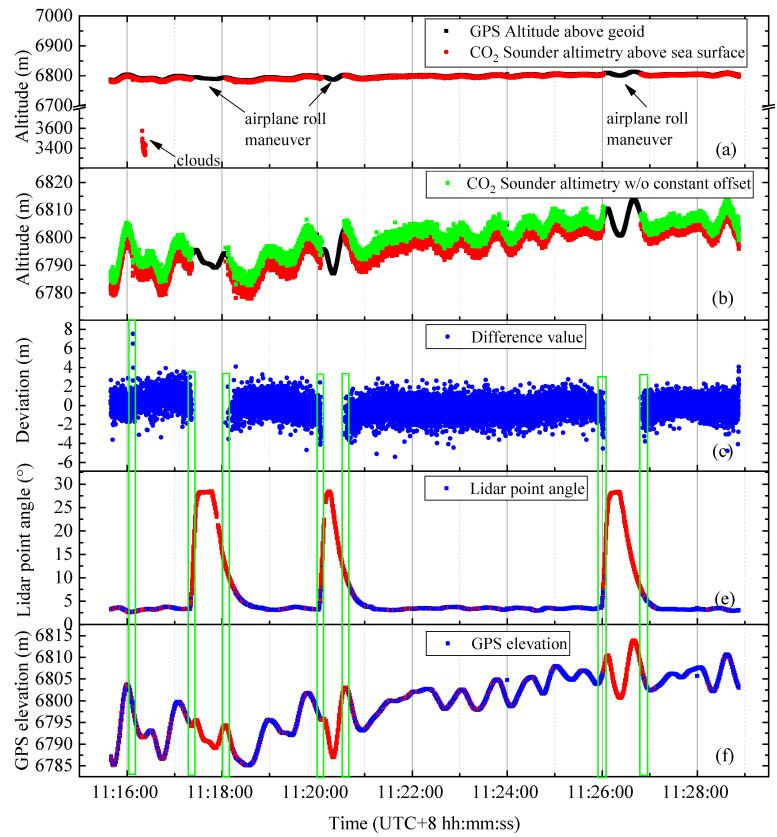
Results of data processing throughout the study area. (**a**) Results of our data processing on the whole sea level research area; the red dots represent the CO_2_ optical column length results from the plane to the sea level, whereas the black dots represent the airborne GPS elevation detection data. The green points in (**b**) consist of two parts lengths. One part is the vertical length calculated by the geometric model after the atmospheric delay correction, and the other part is composed of sea level data from the tide station and the outliers of the single point elevation. The blue points in (**c**) represent the difference between the CO_2_ column length corrected to vertical at each point and the elevation of the airborne GPS. The red points in (**d**–**f**) represent the distribution of Lidar pointing angle and GPS elevation in the whole study area, and the blue points represent the effective data that can be used to invert CO_2_ column length.

**Figure 10 sensors-20-05887-f010:**
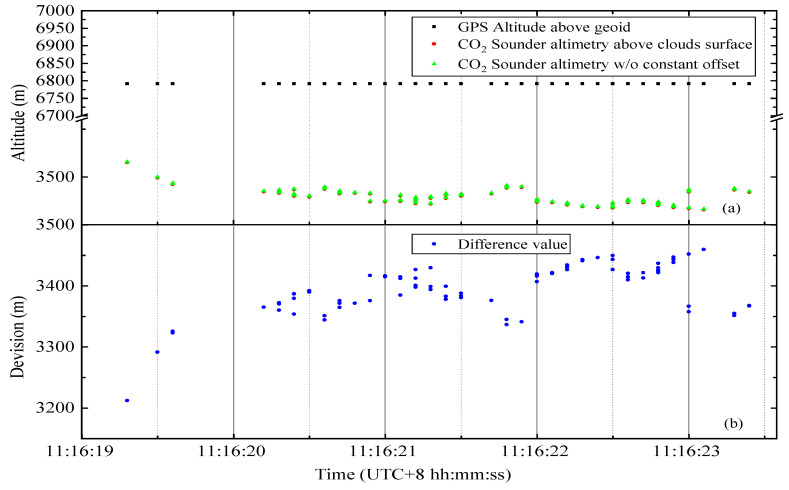
Result of processing a piece of continuous cloud data in the study area. (**a**) The black dots and green and blue points have the same meaning as those in Figure 9. However, the red points represent the CO_2_ column length from the airplane to clouds. (**b**) The blue points represent the difference between the CO_2_ column length corrected to vertical at each point and the elevation of the airborne GPS.

**Table 1 sensors-20-05887-t001:** Parameter values from each formula term.

Parameter Name/Unit	Value	Parameter Name/Unit	Value	Parameter Name /Unit	Value
R $/J\;mol^{-1}K^{-1}$	8.314510	$M_{w}$ / kg/mol	0.018015	$a_{0}$ $/KPa^{-1}$	$1.58123\times 10^{-6}$
$k_{0}/\mu m^{-2}$	238.0185	$w_{0}/\mu m^{-2}$	295.235	$a_{1}/Pa^{-1}$	$-2.9331\times 10^{-8}$
$k_{1}/\mu m^{-2}$	5792105	$w_{1}/\mu m^{-2}$	2.6422	$a_{2}/K^{-1}Pa^{-1}$	$1.1043\times 10^{-10}$
$k_{2}/\mu m^{-2}$	57.362	$w_{2}/\mu m^{-4}$	−0.032380	$b_{0}/KPa^{-1}$	$5.707\times 10^{-6}$
$k_{3}/\mu m^{-2}$	167917	$w_{3}/\mu m^{-6}$	0.004028	$b_{1}/Pa^{-1}$	$-2.051\times 10^{-8}$
cf	1.022	$A/K^{-2}$	$1.2378847\times 10^{-5}$	$c_{0}/KPa^{-1}$	$1.9898\times 10^{-4}$
α	1.00062	$B/K^{-1}$	$-1.9121316\times 10^{-2}$	$c_{1}/Pa^{-1}$	$-2.376\times 10^{-6}$
$\beta /Pa^{-1}$	$3.14\times 10^{-8}$	C	33.93711047	$d/K^{2}Pa^{-2}$	$1.83\times 10^{-11}$
γ	$5.6\times 10^{-7}$	D/K	$-6.3431645\times 10^{3}$	$e/K^{2}Pa^{-2}$	$-0.765\times 10^{-8}$

**Table 2 sensors-20-05887-t002:** Parameters of the used IPDA Lidar.

Parameter Name	Value
Platform height/km	around 6.9
Lidar energy/mJ	6
On-line wavelength/nm	1572.024
Off-line wavelength/nm	1572.085
Repetition frequency/Hz	20 (double pulse)
Pulse width/ns	15–20
Lidar pulse repetition rate/Hz	20
Lidar divergence angle/mrad	0.62
Field angle of the receiver/mrad	1.0
Telescope diameter/m	0.15
Sampling rate/MHz	125
Beam emission interval/us	200

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
