# Peer review of "High-Precision CO2 Column Length Analysis on the Basis of a 1.57-μm Dual-Wavelength IPDA Lidar"

_sensors, 2020, doi:10.3390/s20205887_

Round 1

Reviewer 1 Report

Manuscript ID:  sensors-957704     Type: article
Title: High-Precision CO2 Column Length Analysis on the basis of a 1.57-μm Dual-wavelength IPDA Lidar
Authors: Xin Ma, Haowei Zhang *, Ge Han, Hao Xu, Tianqi Shi, Wei Gong, Yue Ma and Song Li

Dear Dr. Ma:

This paper reports a set of methods on the basis of the actual conditions to accurately measure the exact distance of the Lidar beam to the scattering surface. The methods including autocorrelation  detection, adaptive filtering, Gaussian decomposition, and optimized Levenberg–Marquardt fitting based on the generalized Gaussian distribution. Combined with actual airborne experiment in Qinhuangdao, China on March 14, the author also considered the effect of atmospheric delay on distance measurements. The results showed that the ranging accuracy can reach 0.9066 m, which achieved an excellent ranging accuracy on 1.57 ?m IPDA Lidar, and met the requirement for high precision CO2 column length inversion. This research work is of great significance for improving the accuracy of CO2 concentration inversion by IPDA. However, the manuscript should be revised before acceptance.

Comments:

  1. On line 101, the author said ????2(?) is the weighted integral function of CO2. As far as I know, ????2(?) is just the weighted function and IWF ??2 is the weighted integral function of CO2. IWF is the integral of WF to the pressure.

  1. Line 264 and 265 said that” we can consider the Bohai Sea surface to be relatively calm according to the experimental weather records on that day.” It is recommended to clearly list the weather conditions of the day.

  1. It is recommended to include latitude and longitude information in Figure 3.

  1. Line 288 said that laser pointing angle can be calculated from the pitch and the roll angle, suggest that the author can provide a specific calculation formula.

  1. Figure 9 (c) shows the difference between the CO2 column length corrected to vertical at each point and the elevation of the airborne GPS. I want to know why this difference has positive and negative values?

  1. For high-precision measurements of the CO2 column concentration, the signal-to-noise ratio is very important. As we all know, the ocean's surface reflectivity is very low, resulting in weak power of the echo signal. In this regard, has the author considered how to accurately retrieve the CO2 concentration information?

Author Response

Dear reviewer, thank you for your kindly suggestions,Please see the attachment.

Reviewer 2 Report

The paper describes the idea and improvement of a method to process lidar ranging data to support CO2 column concentration quantification.

The paper is well structured and the reader is nicely guided through the content and, by that, to the authors’ approach. 

For me, there are some interesting and sometimes additional issues to be changed, anyway minor changes of concern: 

+In the introduction the authors are stating in line 34 that a 0.3% uncertainty of the CO2 column concentration would be desirable. However, the best static CO2 concentration reference, i.e. in China, at ambient levels can only be realized with an accuracy of 0.4% rel. 

+The authors try to invest into an Error analysis as a section of its own, section 5. However they do not provide a single quantitative uncertainty assessment of their results.

+ The authors should improve the editing of equation (1): introducing the quantities by their symbols, symbols according to standard conventions like a single letter symbol for a quantity, appropriate subfixes, etc..

+ in line 42 the authors are referring to denominators which haven‘t been explained; in line 131 the full stop after ‘quality’ is missig; 

Author Response

(The authors gave the same response as above.)

Reviewer 3 Report

Comments

Line 203: adjust the apex

Line 216: ‘Ciddor1996’???? adjust the reference…

Lines 236-238, 247-50: it could be much more readable if the values were represented in a table.

Lines 95-96: Your on-line and off-line wavelength choices (Table1) are: 1,572.024 nm and 1,572.085 nm respectively, therefore the difference between the two wavelengths is: 0.061 nm. The FWHM of CO2 peak absorption is about 0.1 nm. It means your off-line wavelength is also attenuated by the CO2 gas. In fact, in figure 4 we can observe this small difference. Is it enough to retrieve the CO2 concentration?

Figure 7: it’s difficult to read with six overlapping curves with six different colors. I suggest to split in two figures separated.

Table 1: with 6 mJ of laser energy and an average altitude of almost 7km it’s a great challenge to collect echo signal from the sea. It could be very interesting to know the features of used detector: responsivity, gain, noise equivalent power…

Suggestions

This paper is well done: The English is very clear. The proposed figures are almost always reported clearly and precisely, also the flowchart representation. Almost always correct and well-referenced bibliography.

Author Response

Dear reviewer, thank you for your kindly suggestions.Please see the attachmen.
